# Analyzing the Interrelationships among Various Ecosystem Services from the Perspective of Ecosystem Service Bundles in Shenyang, China

**Shuang Gan** [1], **Yu Xiao** [2,3,*], **Keyu Qin** [2,3], **Jingya Liu** [2,3], **Jie Xu** [4], **Yangyang Wang** [2,3], **Yingnan Niu** [2,3], **Mengdong Huang** [2,3] and **Gaodi Xie** [2,3]

[1] Graduate School of Media and Governance, Shonan Fujisawa Campus, Keio University, Fujisawa 252-0882, Japan; gans.17s@igsnrr.ac.cn
[2] Institute of Geographic Sciences and Natural Resources Research, Chinese Academy of Sciences, Beijing 100101, China; qinkeyu@igsnrr.ac.cn (K.Q.); liujy.17b@igsnrr.ac.cn (J.L.); wangyy.18b@igsnrr.ac.cn (Y.W.); niuyn.19b@igsnrr.ac.cn (Y.N.); huangmd.19s@igsnrr.ac.cn (M.H.); xiegd@igsnrr.ac.cn (G.X.)
[3] College of Resources and Environment, University of Chinese Academy of Sciences, Beijing 100049, China
[4] School of Ecology and Nature Conservation, Beijing Forestry University, Beijing 100083, China; jiexu@bjfu.edu.cn
[*] Correspondence: xiaoy@igsnrr.ac.cn

**Abstract:** An understanding of the relationships among multiple ecosystem services (ES) facilitates ecosystem management and decision-making. The study of ES bundles can well explain the complex interactions between different ES in a region. Shenyang is a significant economic development and food production area in Northeast China, with a diverse range of ES types. In this study, we quantified eleven ES from Shenyang, China (two provisioning services, eight regulating services, and one cultural service). The trade-offs and synergies among ES were analyzed by Spearman's correlation analysis. The ES bundles were identified using principal component analysis and k-means cluster analysis. Finally, the random forest method was employed to identify the driving factors affecting the ES bundles. The results showed: (1) all ES in Shenyang improved between 2000 and 2019; (2) the most obvious trade-off was found between sand fixation and water conservation; (3) the ES in the study region could be clustered into five different ES bundles which were primarily affected by land-use type; and (4) social-ecological factors largely explained and predicted the formation and distribution of ES bundles. The study provides reference information for the management and optimization of Shenyang's ecosystems and land use regulation.

**Keywords:** ecosystem service bundle; trade-off; synergy; socio-ecological factors; spatial analysis

## 1. Introduction

Ecosystem services (ES) are the benefits that humans derive from ecosystems [1], which include the material resources and services that ecosystems provide to human society for human survival and are closely related to human well-being [2]. The study of ES provides a comprehensive perspective on ecosystem utilization and management, and it is an important tool for quantifying human–nature relationships [3,4]. According to the Millennium Ecosystem Assessment, ES can be classified as provisioning services, regulating services, supporting services, and cultural services [5]. A wide range of ES have complex trade-offs and synergies. For example, the improvement of some provisioning services is frequently accompanied by the reduction of regulating and cultural services [6]. The current ecological and environmental problems are primarily manifested in the simplification of regional ES, which leads to the continuous deterioration of various ES capabilities and the trade-off relationship among various ES and has a significant impact on human health and happiness [7]. Effective management of multiple ecosystems, as well as incorporating

them into landscape management decisions and policy making, necessitates a thorough understanding of the interrelationships among various ES [8–10].

ES bundles are groups of ES that recur in space or time and have similar trade-offs and synergies [8]. It is made up of multiple ES with similar supply levels, trade-offs and synergies. It is produced as a result of the combined effects of natural and social-economic factors, and it can characterize the characteristics of a human–land coupling complex system to some extent. Different types of ES bundles distinguish the human–land coupling complex system, allow the ecological environment problems that exist under different bundle types to be targeted identified, and the internal ES composition, trade-offs, and synergies to be grasped. Propose effective policy measures to avoid or reduce trade-offs based on the trade-offs and synergies, which will help improve the multiple supply of ES and maximize the social well-being of ES [11]. The study of ES bundles aids in understanding the complex interactions among multiple ES, providing guidance for the layout of ecological space as well as the theoretical foundation for effective ecosystem management.

ES bundles are currently being investigated in order to investigate the interaction of regional ES and the relationship between ES spatial distribution and land cover. Crouzat et al., analyzed ES bundles in the French Alps and explored the relationship between ES and land cover [12]. Bai et al., investigated the interaction of ES in Kentucky, identified ES bundles and determined the spatial locations of ES hotspots [13]. According to Kong et al., ES bundles are primarily associated with land use type, slope, and elevation gradient [14]. Jaligot et al., identified ES bundles in the Swiss canton of Vaud and investigated their underlying drivers [15]. Song et al., investigated the trade-offs and synergies among ES in Fuzhou, China, and identified the region's dominant ES types as well as the spatial pattern of ES bundles under different topographic reliefs [16]. Zhang et al., investigated the trade-offs and synergies among ES in China's Yellow River Basin, identified the spatial distribution and main characteristics of ES bundles, and investigated the relationship between ES and land cover [17]. These studies concentrated on analyzing the spatial distribution of ES bundles, comprehending their interrelationships with underlying socio-ecological drivers, and finally, providing a scientific foundation for regional management [18].

Although the research on ES bundles has yielded some results, the majority of them focus on weighing the spatial heterogeneity of the interaction relationship while ignoring the temporal heterogeneity. However, research has revealed that the interaction between ES is not static [9,19]. Understanding the dynamics of ES interactions is essential for effective ES management and avoiding potentially adverse trade-off relationships. Therefore, understanding ES interactions at different time scales is thus another key challenge [20]. Today, both positive (ecological engineering) and negative human activities have a growing impact on the structure and function of ecosystems; thus, profoundly affecting the interactions among ES [21]. Temporal changes in regional ES bundles and their drivers are critical for comprehending ES trade-offs and synergies. This, in turn, helps to improve regional landscape management. However, few studies have focused on the temporal changes in regional ES bundles, as well as the associated trade-offs and synergies among the constituent ES. At the same time, while current research on ES bundles generally focuses on the trade-offs and synergies among ES, the majority of them are analyzed from the research area as a whole, with relatively few studies on the trade-offs and synergies within each identified bundle. It is necessary to conduct additional research in this area.

Shenyang is a significant economic development and grain production area in Northeast China, with a diverse range of ES types. Since 2000, with the implementation of regional ecological restoration projects and the acceleration of urbanization, the disturbance of human activities in this area has continued to increase, and it is urgent to find management solutions that balance economic development and ecological protection. This study takes Shenyang as the research area, and the first step was to conduct a comparative analysis of the trade-offs and synergies among Shenyang's ES as a whole between 2000 and 2019. The ES bundles were then identified at various times, the distribution status and changes of each ES bundle from time and space were analyzed, and the trade-offs and

synergies among ES within each bundle were grasped. Finally, the main driving forces influencing the formation of ES bundles were investigated. This study conducts a dynamic analysis of the interrelationships and driving factors of various ES from the perspective of ES bundles, which is useful in providing a scientific reference for the formulation of relevant ecological protection policies.

## 2. Materials and Methods

### 2.1. Study Area

Shenyang (122°25′9″ E–123°48′24″ E, 41°11′51″ N–43°2′13″ N) is located in the middle of the Liaohe Plain, Liaoning Province. It is surrounded by Fushun City in the east, Benxi City and Liaoyang City in the south, Tai'an and Heishan County in the west, and Kezuohou Banner, Inner Mongolia Autonomous Region in the north. Almost the entire region is dominated by plains, while mountains and hills are concentrated in the northeast. This vast plain has a drainage area of more than 100 km$^2$. A total of 4 large rivers (Liaohe, Hunhe, Raoyang, and Liuhe), four medium-sized rivers, and 18 small rivers flow through the territory. The climate is generally a temperate continental monsoon with four distinct seasons. The mainland cover type is farmland (~70%), while 15% of the city's land area is covered by an artificial surface such as constructed transportation area.

### 2.2. ES Simulation

The selection was made in conjunction with existing literature [22–25] after considering the types of ecosystems in Shenyang and their ecological, economic, and social functions, as well as the availability of data. Shenyang's land use type is primarily farmland, accounting for roughly 70% of total area. As a result, food supply is a significant ES in Shenyang. Shenyang also has forest land distribution in various regions, including deciduous broad-leaved forest, evergreen coniferous forests, deciduous coniferous forests, mixed coniferous and broad-leaved forests, deciduous broad-leaved shrub forests, arbor gardens, shrub gardens, arbor green spaces, etc., the vegetation types are relatively rich, the functions of $SO_2$ absorption, $NO_x$ absorption, carbon sequestration, soil conservation, humidification are relatively prominent. Grassland is distributed in some areas, which can provide fodder for livestock and have the function of grass supply. Shenyang is in China's northeastern region, close to the source of sandstorms, and is part of the Three-North Shelter Forest Project. The sand fixation of forest is very important. The rivers are vertical and horizontal, with the Liaohe, Hunhe, Raoyanghe, and many other rivers, which have the functions of water conservation and flood regulation. The distribution of forests and rivers provides places for outdoor recreation. Finally, we chose 11 ES indicators to study ES bundles. These include two provisioning services (food supply, grass supply), eight regulating services (water conservation, soil conservation, sand fixation, humidification, $SO_2$ absorption, $NO_x$ absorption, flood regulation, carbon sequestration), and one cultural service (outdoor recreation). Based on meteorological data (from the China Meteorological Data Network), Net Primary Productivity (NPP), land cover, and (Normalized Difference Vegetation Index) NDVI data in 2000 and 2019, this study uses ArcGIS (ver. 10.2) to calculate 11 ES. The spatial resolution of each raster data is 30m × 30 m. The specific calculation methods and data sources are shown in Table 1 and Table S6 and the specific calculation methods are given in the Supplementary Materials.

### 2.3. Correlation Analysis of Various ES

To analyze the relationship between individual ES and quantitatively evaluate the trade-offs and synergies between ES, we performed Spearman's correlation analysis in R (ver. 3.4.1). Using the correlation coefficient bubble chart, we then visualized the relationship between the indicators. All significant correlation coefficients were classified as: highly correlated ($|r| >= 0.5$), moderately correlated ($0.3 =< |r| < 0.5$), and weakly correlated ($0.1 =< |r| < 0.3$) according to Cohen 1992 [33].



**Table 1.** Calculation methods and data sources of ecosystem services.

| Category | ES | Unit | Method | Source |
|---|---|---|---|---|
| Provisioning service | Food supply | t/hm$^2$ | Statistical food production in the study area and using the NPP value as the weight to obtain the spatial distribution pattern of food supply [26] | Statistical data comes from the Shenyang Statistical Yearbook, and NPP data comes from NASA |
| | Grass supply | g/m$^2$ | NPP [26] | NASA |
| Regulating service | Water conservation | mm | Water balance [27] | China Meteorological Data Network, MODIS, Shenyang Institute of Applied Ecology, Chinese Academy of Sciences |
| | Carbon sequestration | kg/hm$^2$ | NPP [28] | NASA |
| | SO$_2$ absorption | kg/hm$^2$ | NPP adjusts the purification capacity | NASA |
| | NO$_x$ absorption | kg/hm$^2$ | NPP adjusts the purification capacity | NASA |
| | Humidification | kg/hm$^2$ | LAI [28] | NASA |
| | Soil conservation | t/hm$^2$ | RUSLE [29] | China Meteorological Data Network, MODIS, Shenyang Institute of Applied Ecology, Chinese Academy of Sciences |
| | Flood regulation | m$^3$/hm$^2$ | Calculated based on the area of the water body and the average water depth [30] | Shenyang Institute of Applied Ecology, Chinese Academy of Sciences |
| | Sand fixation | kg/m$^2$ | RWEQ [31] | China Meteorological Data Network, MODIS, Shenyang Institute of Applied Ecology, Chinese Academy of Sciences |
| Cultural service | Outdoor recreation | Index | Weighted calculation based on recreational opportunities provided by three ecosystem landscape indicators [32] (environmental naturalness, number of land cover types, and surface roughness) | MODIS, Shenyang Institute of Applied Ecology, Chinese Academy of Sciences |

*2.4. Identification of ES Bundles*

We identified ES bundles in three steps. In the first step, we used principal component analysis (PCA) (in R 3.4.1) of 11 ES to identify principal components to reduce the dimensionality of the data and reduce some redundant variables. The principle of PCA is to try to recombine the original variables with a certain correlation to obtain a new set of comprehensive variables that are independent of each other but reflect the original variable information as much as possible. Cluster analysis of the principal components obtained by PCA can help to obtain more reliable clustering results [34]. In the second step, we used the silhouette coefficient to determine the optimal number of ES bundles. The value range of the silhouette coefficient is ($-1$, 1), and the closer to 1, the better the clustering effect. In the

third step, based on the principal components and optimal bundle numbers obtained from the first two steps, K-means cluster analysis was used to quantify the ES bundles. Finally, we used ArcGIS (ver. 10.2) to spatially visualize these ES bundles and draw a rose diagram for each ES Bundle to identify the spatial interactions among them.

### 2.5. Identifying Drivers That Affect ES Bundles

There are certain differences among the ES bundles. The driving factors for the formation of ES bundles were identified through the spatial distribution of some social-ecological indicators. These indicators, which were mainly obtained by summarizing existing literature [35–37], include 10 indicators: namely, land use (forest, grassland, farmland, water, urban construction land), annual precipitation, annual average temperature, average wind speed, elevation, and population density. We used R to perform random forest analysis, a compositional supervised learning method for the abovementioned 10 indicators. Random forest refers to a classifier that uses multiple trees to train and predict samples. It can provide stable results for correlated predictor variables and allows assessing their relative importance [38]. We firstly used 70% random samples to analyze the importance of 10 socio-ecological indicators. Then, we used the remaining 30% to predict the ES bundles in the study area to evaluate the identification accuracy of the driving factors. Finally, we used ArcGIS to spatialize the predicted ES bundles.

## 3. Results

### 3.1. Temporal and Spatial Changes in ES

Our results indicated that (Figure 1), except for the grass supply, sand fixation, and flood regulation, other ES in Shenyang are higher in the southeastern municipal districts. The grass supply was mainly related to the distribution of grasslands, largely located in the northern and central regions. The sand fixation was primarily affected by wind speed, with a decreasing trend from west to east. While the flood regulation was mainly related to the distribution of water bodies with the highest concentration in the northern area.

We analyzed the changes in Shenyang's ES over the past 20 years by calculating the difference in the amount of each ES in 2019 compared to 2000. Except for the $SO_2$ absorption, $NO_x$ absorption and the sand fixation, each ES had increased in 2019 in more than 90% of the study area (Figure 2, Table S7). In contrast, only 52% of the area showed an increase in the amount of $SO_2$ absorption and $NO_x$ absorption. Sand fixation increased in less than 2% of the areas, while more than 98% of the area showed a decreasing trend.

### 3.2. Trade-Offs and Synergies of ES

Correlation between ES in Shenyang varied year-wise: Among the 55 pairs of ES, 55 pairs of ES were correlated in 2000 while 52 pairs in 2019. In 2000, there were 9 highly correlated ($|r| \geq 0.5$), 8 moderately correlated ($0.3 \leq |r| < 0.5$), and 22 weakly correlated ($0.1 \leq |r| < 0.3$) ES pairs whereas, there were 8 highly correlated, 5 moderately correlated and 23 weakly correlated ES pairs in 2019 (Figure 3). In 2000, there were 15 pairs of synergistic relations and 2 pairs of trade-off relations among the 17 pairs of highly and moderately correlated relations. Food supply, carbon sequestration, $SO_2$ absorption, $NO_x$ absorption, and outdoor recreation had high synergies with various ES. Among them, the synergy between $SO_2$ absorption and $NO_x$ absorption was the highest, with a correlation coefficient of 0.99. Carbon sequestration and food supply, carbon sequestration and $SO_2$ absorption, carbon sequestration and $NO_x$ absorption, food supply and $SO_2$ absorption, food supply and $NO_x$ absorption, outdoor recreation and $SO_2$ absorption, outdoor recreation and $NO_x$ absorption were highly correlated. Sand fixation had an obvious trade-off with other ES, and it was highly negatively correlated with water conservation (correlation coefficient of $-0.61$). In addition, sand fixation also had an obvious trade-off with the food supply with a moderate negative correlation between the two. Grass supply, soil conservation, humidification, and flood regulation were not significantly correlated with other ES. In 2019, the trade-off among various ES in Shenyang decreased to 1 pair

compared to those in 2000. Among them, the trade-offs between sand fixation and other ES were significantly reduced. Except for the trade-offs with water conservation, there was no obvious correlation between soil fixation and other ES.

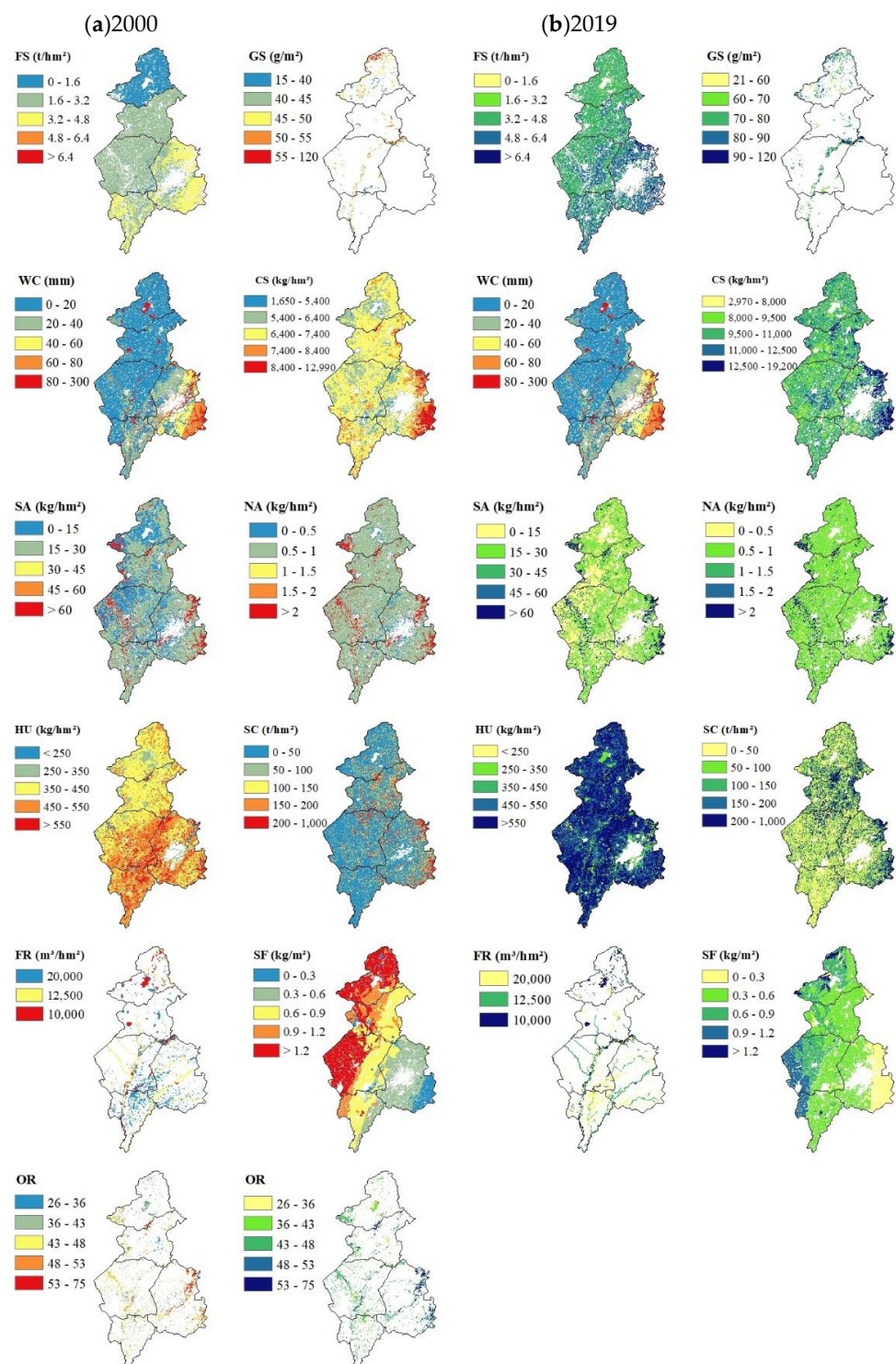

**Figure 1.** Spatial distribution of ES in Shenyang during (**a**) 2000 and (**b**) 2019, food supply (FS), grass supply (GS), water conservation (WC), carbon sequestration (CS), SO$_2$ absorption (SA), NO$_x$ absorption (NA), humidification (HU), soil conservation (SC), flood regulation (FR), soil fixation (SF), outdoor recreation (OR).

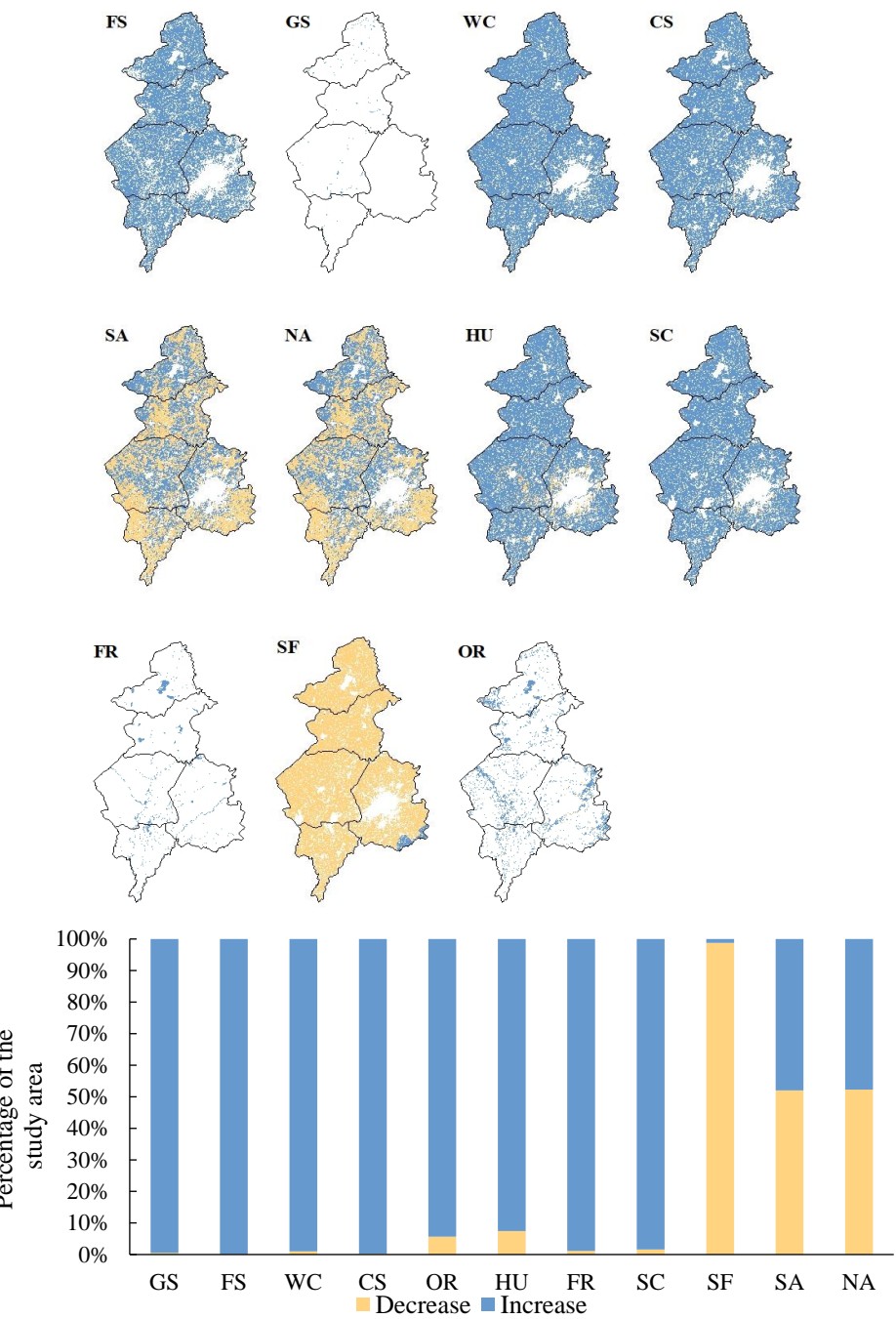

**Figure 2.** Spatial distribution of increases and decreases in ES during 2000 and 2019, food supply (FS), grass supply (GS), water conservation (WC), carbon sequestration (CS), SO$_2$ absorption (SA), NO$_x$ absorption (NA), humidification (HU), soil conservation (SC), flood regulation (FR), soil fixation (SF), and outdoor recreation (OR).

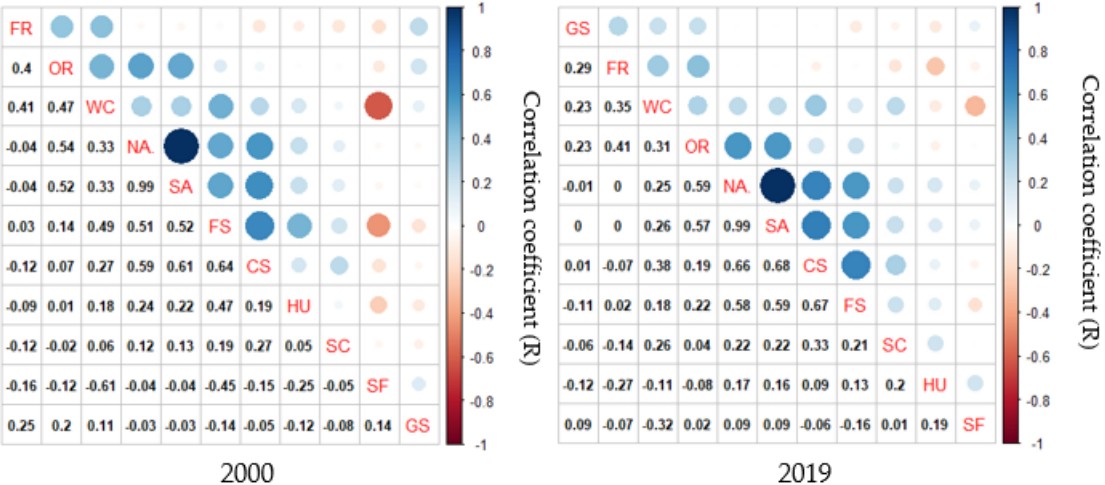

**Figure 3.** Correlation map of ES in Shenyang during 2000 and 2019, food supply (FS), grass supply (GS), water conservation (WC), carbon sequestration (CS), SO2 absorption (SA), NOx absorption (NA), humidification (HU), soil conservation (SC), flood regulation (FR), soil fixation (SF), outdoor recreation (OR). The blue and red circle indicate positive and negative correlations, respectively.

### 3.3. Identification of ES Bundles

In the study region (Shenyang), each ES was counted according to a 500 m × 500 m grid, and accordingly, we obtained data for 50,381 grids. Shenyang was divided into five ES bundles by principal component analysis (Tables S8 and S9), silhouette coefficient and k-means cluster analysis (Table S10). The spatial distribution of each ES bundle is shown in Figure 4.

As shown in Figure 5, the contributions of each ES in ES bundle 1 were not significantly different, with humidification contributing the most, followed by carbon sequestration and food supply, and flood regulation contributing the least. It covered the entire city of Shenyang and was the most widely distributed of the five bundles, accounting for 69.91% of the total area in 2000 and 69.33% in 2019. The functions of each ES in this bundle were relatively weak in comparison to the other four bundles, only slightly higher than bundle 3. Except for the decrease in grass supply from 2000 to 2019, all other ES have improved over time, namely food supply, soil conservation, carbon sequestration, SO$_2$ absorption, NO$_x$ absorption, humidification, sand fixation, flood regulation, water conservation, and outdoor recreation had a synergistic relationship, while they had a trade-off relationship with grass supply. The functions of flood regulation, water conservation, and outdoor recreation were more significant in ES bundle 2, which were the highest of the five bundles, and flood regulation contributes the most. This bundle was mainly distributed in the central and southern parts of Shenyang near water bodies, accounting for 13.55% and 15.45% of total area in 2000 and 2019, respectively, with an increasing trend, mainly converted from bundle 5. Grass supply, NO$_x$ absorption, and sand fixation increased in 2019 compared to 2000, while other ES decreased, indicating that grass supply, NO$_x$ absorption, and sand fixation were synergies and trade-offs with food supply, soil conservation, carbon sequestration, SO$_2$ absorption, humidification, flood regulation, water conservation, and outdoor recreation. Overall, among the five ES bundles, ES bundle 3 had the weakest ES function. It was mainly distributed in Shenyang's southeastern and northern regions, accounting for 4.60% and 5.52% of the total area, respectively. Water conservation contributed the most in 2000, while grass supply contributed the most in 2019. Only carbon sequestration increased from 2000 to 2019, while other ES decreased; that is, each ES in bundle 3 decreased with the increase in carbon sequestration, indicating that they had a trade-off relationship with carbon sequestration. Among the five bundles, ES bundle 4 had the strongest ES function, with SO$_2$ absorption and NO$_x$ absorption contributing the most and sand fixation contributing the least. Mainly distributed in the southeastern and northern parts of Shenyang, accounting

for 7.60% and 7.89% of the total area in 2000 and 2019, respectively. Compared to 2000, in 2019, grass supply, soil conservation, humidification, sand fixation, flood regulation, and outdoor recreation have increased, while others had decreased, namely grass supply, soil conservation, humidification, sand fixation, flood regulation and outdoor recreation were synergistic, while food supply, carbon sequestration, $SO_2$ absorption, $NO_x$ absorption, water conservation were trade-offs. In ES bundle 5, sand fixation and grass supply were the most significant, and sand fixation was the most powerful of the five bundles. Bundle 5 mainly distributed in the northern part of the study area, accounting for 4.33% of the total area in 2000 and 1.80% in 2019. Food supply, carbon sequestration, $SO_2$ absorption, $NO_x$ absorption, humidification, sand fixation, and water conservation all improved in 2019 compared to 2000, while others decreased. Specifically, food supply, carbon sequestration, $SO_2$ absorption, $NO_x$ absorption, humidification, sand fixation, and water conservation are synergies, while they were trade-offs with grass supply, soil conservation, flood regulation, and outdoor recreation (Figures 4 and 5).

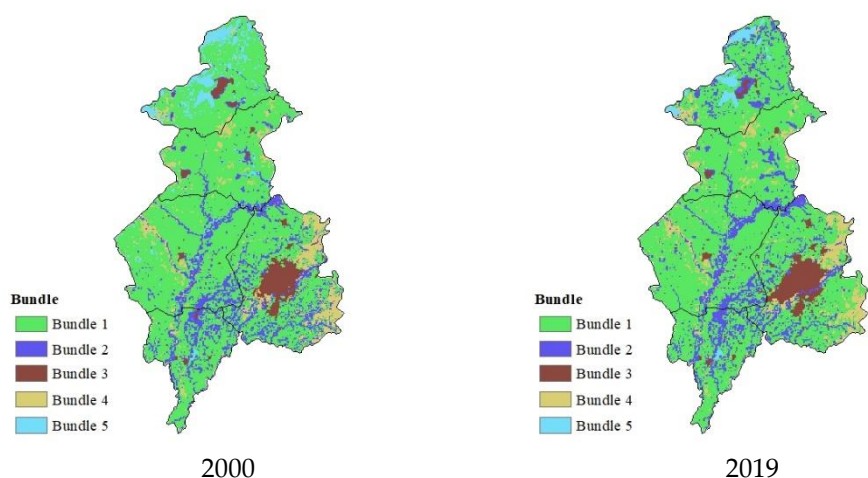

**Figure 4.** Spatial distribution of ES bundles in Shenyang.

### 3.4. Identification of ES Bundles Drivers

Selected relevant socio-ecological factors (e.g., annual average temperature, annual precipitation, annual average wind speed, elevation, forest area, grassland area, water area, farmland area, urban construction land area, and population density) were analyzed to identify the driving factors of ES bundles in Shenyang. The data source of these factors is shown in Table S11. As per the random forest method coupled with the ArcGIS vector map (with 50,381 grids in total), the main driving factors of the ES bundles were identified, and the results are shown in Figure 6. The mean decrease in Gini represents the reduction in the Gini coefficient (the probability that a randomly selected sample in the sample set is misclassified) after variable replacement (the larger the value, the more important is the variable). In 2000 and 2019, the accuracy rates of the random forest analysis on the driving factors of ES bundles were 92.36% and 93.03%, respectively (Tables S12 and S13). The water area was found to be the most important factor. In addition to the influence of water area, annual precipitation and elevation also affected the distribution of ES bundles to a certain extent in 2000, while in 2019, the significant additional factors were the farmland area and elevation (Figure 6).

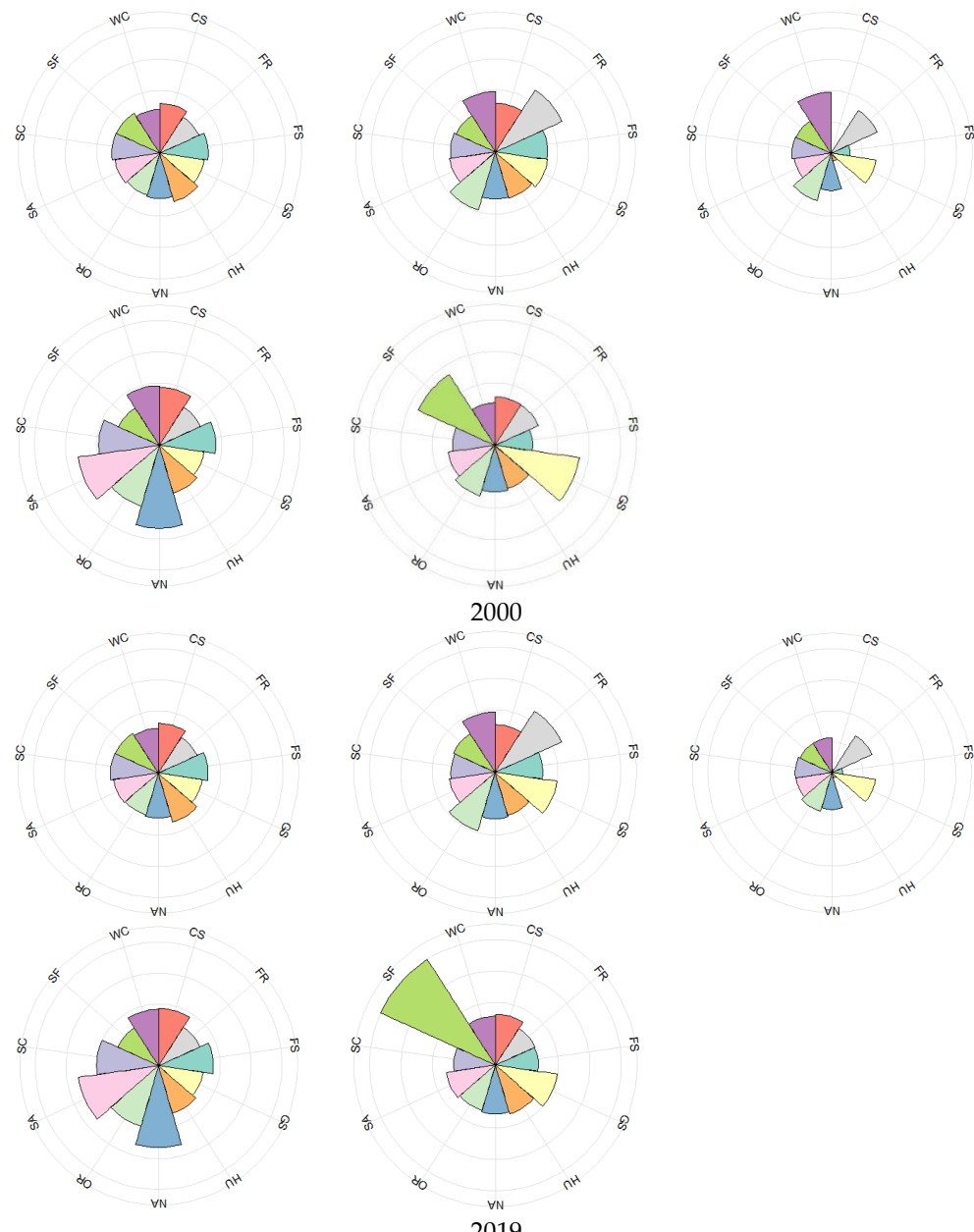

**Figure 5.** Distribution of ES in each ES bundle in Shenyang, food supply (FS), grass supply (GS), water conservation (WC), carbon sequestration (CS), SO2 absorption (SA), NOx absorption (NA), humidification (HU), soil conservation (SC), flood regulation (FR), soil fixation (SF), outdoor recreation (OR).

Moreover, we calculated the prediction accuracy of the driving factors on the remaining 30% of the 50,381 grids, and the results are shown in Figures 7 and 8. The prediction accuracy in 2000 was 92.68%, and the inaccurate parts were mainly concentrated in the southeastern and northern areas of the study area. Bundle 1 had the highest accuracy, while bundle 4 had the lowest. The prediction accuracy in 2019 was 92.82%, which showed an improvement over 2000. The inaccurate regions were mainly concentrated in the western and northern parts of the study area (Figures 7 and 8), and bundle 3 is the highest and bundle 4 is the lowest.

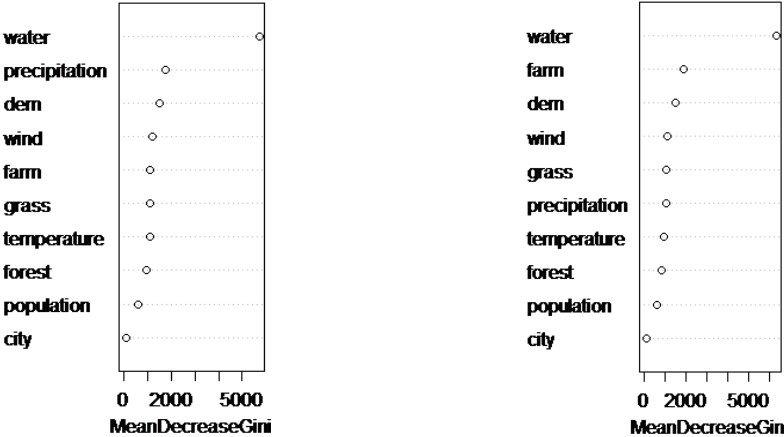

**Figure 6.** Importance of socio-ecological variables.

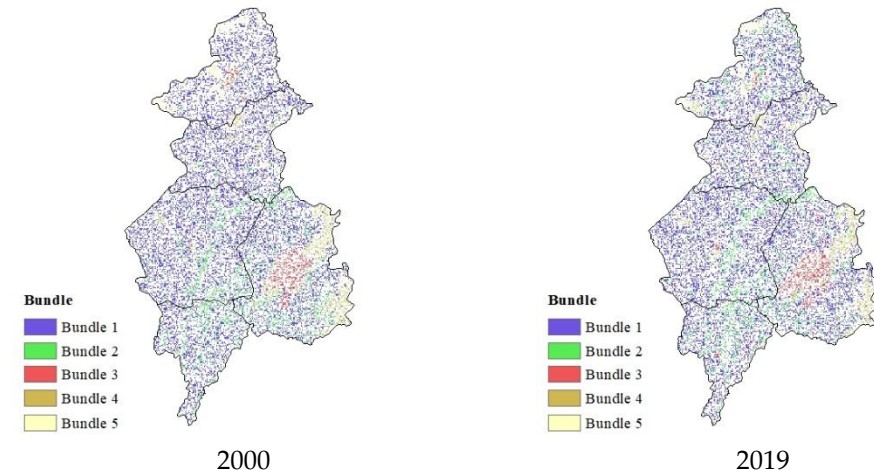

**Figure 7.** Predicted ES bundles.

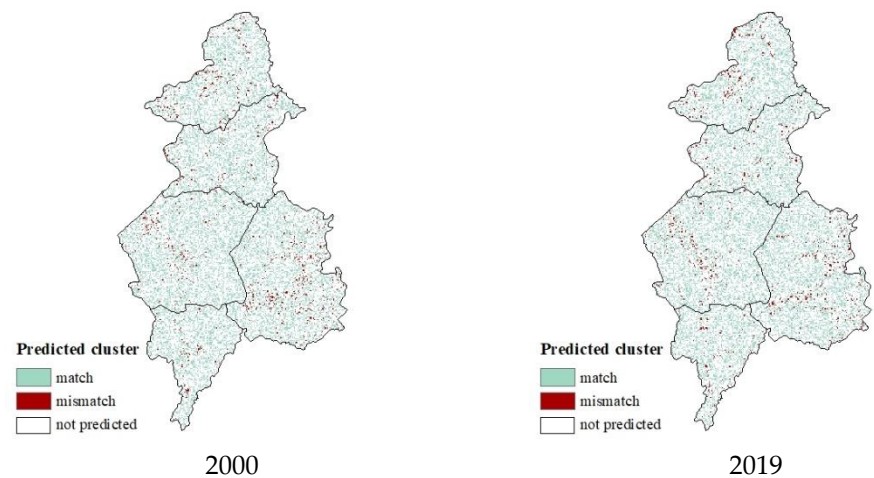

**Figure 8.** Accuracy of predicted ES bundles.

## 4. Discussion

This study provides an important basis for the management and decision-making of regional land use by analyzing the relationship among various ES in Shenyang after evaluating the ES bundles and their driving factors.

### 4.1. Temporal and Spatial Changes in ES

The implementation of ecological engineering has a certain positive effect on the improvement of ES [39,40]. Compared within the mentioned timeframe (2000–2019), sand fixation decreased as a whole, while the rest of the ES showed an increasing trend. During the period from 2000 to 2019, Shenyang implemented the project of returning farmland to the forest, such as the Three North Shelter Forest Project, and so on. The vegetation coverage as well as ES, therefore, have increased significantly [41]. Sustainable development strategies such as agricultural land consolidation and watershed improvement can ensure continued growth in agricultural production even in the context of large-scale restoration and afforestation and significant reductions in agricultural land use [42]. In Shenyang, despite a substantial reduction in the agricultural area, food supply increased due to increasing per unit area yield of grain [43]. The direct effect of shelterbelts on land desertification prevention is mainly to weaken the driving force. By strengthening large-scale restoration and afforestation, the surface wind speed can be weakened, resulting in an obvious wind protection effect. After afforestation in Naiman Woodland, the average wind speed decreased by about 20% (data source and Shenyang Ecological Environment Bureau). Under the influence of the Three-North Shelterbelt Project, the annual average wind speed in Shenyang in 2019 was greatly reduced compared to that in 2000. The estimation of sand fixation is mainly obtained by the calculation of wind erosion. The reduction of annual average wind speed directly affects wind erosion. Therefore, compared with 2000, the sand fixation in Shenyang decreased in 2019.

### 4.2. ES Trade-Offs and Synergies

Water conservation, sand fixation, soil conservation, grass supply, flood regulation, and humidification had weak correlations with other ES. Strong synergistic relationships existed among food supply, carbon sequestration, $SO_2$ absorption, $NO_x$ absorption, and outdoor recreation, this result is consistent with existing research results [40,44,45]. The trade-offs between ES in the study area were mainly related to sand fixation which is consistent with the findings of Niu et al. [46], and water conservation had the most obvious trade-off with sand fixation. The formation of trade-offs and synergies among ES are mainly influenced by the conflict or consistency of land-use types [47]. The trade-off between sand fixation and water conservation was mainly affected by vegetation coverage. The higher the vegetation coverage, the stronger the water conservation. However, the wind speed was lower in places with more vegetation, and the smaller the wind speed, the lower was the sand fixation. This formed a basis for the trade-off relationship. In most of the studies on the trade-offs and synergies of ES, trade-off relationships between provisioning services and regulating services were reported [28,48–50]. A study on the Loess Plateau reported a synergistic relationship between agricultural provisioning and carbon sequestration [40], which is different from the studies that investigated the relationship between agricultural production and increase in forest area [51]. Food supply and carbon sequestration, food supply and $SO_2$ absorption, food supply and $NO_x$ absorption, food supply and water conservation are all synergies. Therefore, to an extent, it can be proved that technological progress and policy support have mitigated the adverse effects of vegetation restoration on agricultural production.

Change in land use is the most important factor that affects ES and changes ES trade-offs and synergies [52,53]. Driven by various ecological projects, during the study period (from 2000 to 2019), most of Shenyang's ES showed a trend that synergies became stronger and trade-offs became weaker. This pattern suits the ecological management goal of strengthening the coordination and reducing the conflicts in ES and thus promotes coordinated development of the ecological environment.

### 4.3. ES Bundles

In this study, we recognized the ES bundle in Shenyang based on a 500 m × 500 m grid instead of administrative regions. Mainly due to the small sample size for identifying ES

bundles based on administrative boundaries, the results are inaccurate and rough in small-scale regions. However, the sample size based on the 500 m × 500 m grid is sufficient, so that the results are more accurate and the detailed distribution of each bundle can be obtained.

There are not only pairwise interactions among ES but also consistent multi-groups, that is ES bundles [48]. ES bundles provide valuable information on the relationships between multiple ES that can be used to develop management strategies [8]. The ES of Shenyang was divided into 5 bundles by k-means cluster analysis. There is a close relationship between ES bundle types and land use types [54], and the distribution of land use can partially explain the distribution of ES bundle [18,44]. The five bundles in Shenyang showed different characteristics affected by different land-use types.

The main land-use type in bundle 1 was farmland, so food supply played an important role in this bundle. Because of the single vegetation type, the contributions of each ES are similar. The area of bundle 1 decreased from 2000 to 2019, mainly due to the implementation of farmland conversion to forests during this time period, which resulted in an increase in forest and a decrease in farmland (Figure S1). The increased forest area enables the bundle to provide better regulating and cultural services. At the same time, the contribution of food supply has also increased, Shenyang increased grain production by increasing the per unit grain yield, so the proportion of food supply in bundle 1 increased despite a decrease in farmland area. Bundle 1's primary function was food supply. As a result, for ecosystem management, the food supply function must be ensured, and each ES other than grass supply must be improved based on the trade-offs and synergies of this bundle. Bundle 2 was mainly distributed around water bodies, making it the most affected by water and allowing the ES that was closely related to water bodies to play the most prominent role in this bundle. The area of this bundle increased from 2000 to 2019, owing primarily to the more widespread distribution of water bodies in the study area (Figure S1). The main function of this bundle was flood regulation. In the process of ecosystem management, planning and decision-making should be focused on flood regulation. Food supply, soil conservation, carbon sequestration, $SO_2$ absorption, humidification, water conservation, and outdoor recreation must all be improved concurrently in planning decision-making by combining the trade-offs and synergies of this bundle. Bundle 3 was mainly distributed in Shenyang's urban area. Because this bundle's land-use type was primarily urban construction land with little vegetation, the effect of each ES did not differ significantly, and each ES of the bundle was the weakest of the five bundles. In 2019, the area of bundle 3 increased in comparison to 2000, owing primarily to a significant increase in urban construction in Shenyang in tandem with economic development, which also results in an overall reduction of each ES in the bundle. Strengthening urban greening and paying attention to adjusting the trade-offs between carbon sequestration and other ES are highly recommended for this bundle's ecosystem management. From 2000 to 2019, the ES bundle 4 area increased slightly, owing primarily to an increase in forest area (Figure S1). $SO_2$ absorption and $NO_x$ absorption were most obvious in bundle 4. The land-use type in this bundle was mainly forest land, which had the strongest ability to absorb $SO_2$ and $NO_x$. As a result, the primary task of this bundle in the process of ecosystem management is to increase forest area to improve $SO_2$ absorption and $NO_x$ absorption, while also combining the trade-offs and synergies of this bundle to improve food supply, carbon sequestration, and water conservation. The most obvious in bundle 5 was sand fixation. It was mainly located in the northwest of Shenyang, which was the closest entry point for wind. As a result, the primary management task for this bundle is further strengthening sand fixation, while improving food supply, carbon sequestration, $SO_2$ absorption, $NO_x$ absorption, humidification, and water conservation.

### 4.4. Identification of ES Bundles Drivers

Raudsepp-Hearne et al., suggested that spatially explicit analyses of the social-ecological variables driving ES bundles could ultimately allow for the prediction and modeling of ES bundles and thus, critical trade-offs and synergies across regions [8]. Even the problems associated with the data-intensive models required to produce ES maps can

be overcome if the prediction of the ES bundle is achieved through extensive access to data on socio-ecological drivers [36]. At present, the main driving factor analysis and ES bundles prediction methods include multiple logistic regression [21,55], random forest analysis method [36,44], redundancy analysis [56–58], and binomial logistic regression [59]. Random forests are generally more accurate than other classification methods. In addition, it can handle multi-variable problems, and can calculate out-of-bag prediction errors and measure variable importance [60]. Therefore, the random forest algorithm was selected in this study to analyze the driving factors of ES bundles in Shenyang.

Land-use change can drive demand and supply in one or more ecosystems [61], so land use is considered a driver of ES bundles in many studies [18,35,59]. Elevation is also a major factor affecting local climate, land use distribution, and ES [62]. Climate factors affect ES by affecting the growth of vegetation. Population density is also an important predictor of ES [18,44]. On this basis, the present study selected the annual average temperature, annual precipitation, annual average wind speed, elevation, forest area, grassland area, water area, farmland area, urban construction land area, and population density as socio-ecological drivers affecting the formation of ES bundles.

*4.5. Limitations of the Study*

The ES selected in this study are mainly regulating services. The indicators of provisioning services are only food supply and grass supply, which cannot reflect the trade-offs and synergies between provisioning services and regulating services comprehensively. Some ES have strong synergies due to the use of the same indicators in the calculation methods. Again, similar calculation methods might affect the results. For example, the calculation methods and required indicators of $SO_2$ absorption and $NO_x$ absorption are similar. Therefore, the results would show a high degree of synergy between them. Hence, more specific metrics may be used to define and quantify the ES for more reliable results.

**5. Conclusions**

This study analyzed the interaction relationships among various ES in Shenyang, identified ES bundles and the driving factors that affect ES bundles. The results indicated that the ES in Shenyang showed an overall increasing trend except for sand fixation, $SO_2$ absorption, and $NO_x$ absorption from 2000 to 2019. Food supply, carbon sequestration, $SO_2$ absorption, $NO_x$ absorption, and outdoor recreation had high synergies with other ES. The trade-offs between ES in the study area were mainly related to sand fixation, and the most obvious trade-off with sand fixation was water conservation. The ES in Shenyang were divided into five ES bundles. Each bundle had different characteristics affected by land-use type. The study also showed that socio-ecological factors largely explained and predicted the formation and distribution of the ES bundles. To manage the ecosystems and make a comprehensive decision support system, future research should combine ES supply with flow and demand to analyze the relationships among various ES.

**Supplementary Materials:** The following supporting information can be downloaded at: https://www.mdpi.com/article/10.3390/land11040515/s1, Table S1: Calculation parameters of stock carrying capacity of different grassland types in Shenyang; Table S2: Mean values of surface runoff coefficients of various ecosystems types; Table S3: Reference value of parameter $\omega$ for evaluating the importance of water conservation function; Table S4: The ability of vegetation to absorb air pollutants; Figure S1: Types of land use in Shenyang; Table S6: Data sources; Table S7: Statistics of ES in Shenyang in 2000 and 2019; Table S8: Loadings of ES onto each principal component in 2000; Table S9: Loadings of ES onto each principal component in 2019; Table S10: The number and proportion of grids per bundle; Table S11: Data source of social-ecological factors; Table S12: Distribution of predicted membership in 2000 (Grid); Table S13: Distribution of predicted membership in 2019 (Grid) [63–66].

**Author Contributions:** S.G., conceptualization, methodology, visualization, writing—original draft and writing—review and editing. Y.X., funding acquisition, supervision, conceptualization, methodology and writing—review and editing. K.Q., writing— review and editing. J.L., writing— review and editing. J.X., writing—review and editing. Y.W., writing—review and editing. Y.N., writing—review

and editing. M.H., writing—review and editing. G.X., writing—review and editing. All authors have read and agreed to the published version of the manuscript.

**Funding:** This research was funded by the National Natural Science Foundation of China (41971272), the Strategic Priority Research Program of Chinese Academy of Sciences (XDA20020402), Guangxi Science and Technology Major Project (AA20161002-3).

**Institutional Review Board Statement:** Not applicable.

**Informed Consent Statement:** Not applicable.

**Data Availability Statement:** Not applicable.

**Conflicts of Interest:** The authors declare no conflict of interest.

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
