# Peer review of "Analyzing the Interrelationships among Various Ecosystem Services from the Perspective of Ecosystem Service Bundles in Shenyang, China"

_land, doi:10.3390/land11040515_

Round 1

Reviewer 1 Report

Thank you very much for the interesting article. Ecosystem services (ES) provision and respective trade-offs between various ES is an important research topic.

Introduction should outline the problem and summarize the state of the art. This is done very briefly. Why is this topic relevant for investigation? What is the situation in China? How was the research gap identified? Are there similar studies in China? Please explain further.

Why did you decided to choose 500m × 500m grid instead of administrative regions? Please specify.

Material and methods

Lines 80_90: “Based on the natural and social conditions of the study area in combination with existing literature, the present study selected 11 ES indicators to study ES bundles.”  This is very general. The methodology should ensure replicability. How were the ES indicators for bundles selected? Based on which natural and social conditions? What literature was used? This needs to be specified further. Why only two provisioning, eight regulating and one cultural?

In section 2.5 you clearly identified sources from which the indicators were taken (references 23-25). Please do the same for section 2.2.

Results are presented according to the aims. Section 3.2 is very difficult to follow, it contains too much data and indicators. From figure 3 it is not clear what the blue and red colour means, because the legend is missing.

Discussion is sufficient and conclusions are supported by results.

Overall, the article presents a valuable contribution to the ES provision debate.

Author Response

Thank you for your comments concerning our manuscript. Those comments are all valuable and very helpful for revising and improving our paper, as well as the important guiding significance to our researches. We have studied comments carefully and have made correction which we hope meet with approval. The main corrections in the paper and the responds to the reviewer’s comments are as flowing:

Point 1: Introduction should outline the problem and summarize the state of the art. This is done very briefly. Why is this topic relevant for investigation? What is the situation in China? How was the research gap identified? Are there similar studies in China? Please explain further.

Response 1: Thank you very much for your good comment. The introduction does have a lot of deficiencies, and we have made an overall revision of the introduction according to your suggestion.

Point 2: Why did you decided to choose 500m × 500m grid instead of administrative regions? Please specify.

Response 2: Thank you very much for your good comment. The specific reason has been added in 4.3. Mainly due to the small sample size for identifying ES bundles based on administrative boundaries, the results are inaccurate and rough in small-scale regions. According to 500m × 500m grid calculation, the study area can be counted as 50381 grids, and the sample size is large enough, which makes the results more accurate, and the detailed distribution of each bundle can be obtained.

Point 3: Lines 80_90: “Based on the natural and social conditions of the study area in combination with existing literature, the present study selected 11 ES indicators to study ES bundles.”  This is very general. The methodology should ensure replicability. How were the ES indicators for bundles selected? Based on which natural and social conditions? What literature was used? This needs to be specified further. Why only two provisioning, eight regulating and one cultural?

Response 3: Thank you very much for your good comment. We have added specific reasons for choosing 11 ES indicators in Material and methods. The selection was made in conjunction with existing literature after considering the types of ecosystems in Shenyang and their ecological, economic, and social func-tions, as well as the availability of data. Shenyang's land use type is primarily farmland, accounting for roughly 70% of total area. As a result, food supply is a significant ES in Shenyang. Shenyang also has forest land distribution in various regions, including de-ciduous broad-leaved forest, evergreen coniferous forests, deciduous coniferous forests, mixed coniferous and broad-leaved forests, deciduous broad-leaved shrub forests, ar-bor gardens, shrub gardens, arbor green spaces, etc., the vegetation types are relatively rich, the functions of SO2 absorption, NOx absorption, carbon sequestration, soil con-servation, humidification are relatively prominent. Grassland is distributed in some areas, which can provide fodder for livestock and have the function of grass supply. Shenyang is in China's northeastern region, close to the source of sandstorms, and is part of the Three-North Shelter Forest Project. The sand fixation of forest is very im-portant. The rivers are vertical and horizontal, with the Liaohe, Hunhe, Raoyanghe, and many other rivers, which have the functions of water conservation and flood reg-ulation. The distribution of forests and rivers provides places for outdoor recreation. Finally, we chose 11 ES indicators to study ES bundles.

Point 4: In section 2.5 you clearly identified sources from which the indicators were taken (references 23-25). Please do the same for section 2.2.

Response 4: Thank you very much for your good comment. We have added references for selecting indicators in Material and methods.

Point 5: Results are presented according to the aims. Section 3.2 is very difficult to follow, it contains too much data and indicators. From figure 3 it is not clear what the blue and red colour means, because the legend is missing.

Response 5: Thank you very much for your good comment. We have modified Figure 3, added legends, added the full name of each abbreviation and the meaning represented by the blue and red circles in the title section.

Thank you very much for your comments and suggestions again.

Sincerely,

Shuang Gan

Reviewer 2 Report

A very interesting and excellently presented manuscript. The only things that I missed were some more elaboration on the environmental part of the analysis e.g.

- on the  actual linkages between policy interventions and the relative change in ES provision, 

- about the plausibility of ES trade offs and synergies identified 

Some comments on the temporal variations 2000-2019  within ES bundles.

Furthermore, some very small language problems that can be solved by a thorough reading. 

 I would like to congatulate the authors.

Author Response

Thank you for your comments concerning our manuscript. Those comments are all valuable and very helpful for revising and improving our paper, as well as the important guiding significance to our researches. We have studied comments carefully and have made correction which we hope meet with approval. The main corrections in the paper and the responds to the reviewer’s comments are as flowing:

A very interesting and excellently presented manuscript. The only things that I missed were some more elaboration on the environmental part of the analysis e.g.

Point 1: on the actual linkages between policy interventions and the relative change in ES provision,

Response 1: Thank you very much for your good comment. The actual linkages between policy interventions and the relative change in ES provision is described in Section 4.3. From 2000 to 2019, the overall trend of ES in Shenyang increased, mainly due to the influence of ecological engineering. During this period, Shenyang implemented the project of returning farmland to forest, the Three North Shelter Forest Project and other projects, the vegetation coverage has increased significantly, thus making the ES increase. At the same time, we also added corresponding policy recommendations based on the leading ES and the trade-offs and synergies among ES of each bundle in Section 4.3. For example, bundle 1's primary function was food supply. As a result, for ecosystem management, the food supply function must be ensured, and each ES other than grass supply must be improved based on the trade-offs and synergies of this bundle. The main function of bundle 2 was flood regulation. In the process of ecosystem management, planning and decision-making should be focused on flood regulation. And food supply, soil conservation, carbon sequestration, SO2 absorption, humidification, water conservation, and outdoor recreation must all be improved concurrently in planning decision-making by combining the trade-offs and synergies of this bundle. In terms of ecosystem management of bundle 3, strengthening urban greening and paying attention to adjusting the trade-offs among carbon sequestration and other ES are highly recommended. Bundle 4 was dominated by absorb SO2 and NOx. Therefore, the primary task of this bundle in the process of ecosystem management is to increase forest area to improve SO2 absorption and NOx absorption, while also combining the trade-offs and synergies of this bundle to improve food supply, carbon sequestration, and water conservation. The dominant ES of bundle5 is sand fixation, the main management task for this bundle is further strengthening sand fixation, while improving food supply, carbon sequestration, SO2 absorption, NOx absorption, humidification and water conservation.

Point 2: about the plausibility of ES tradeoffs and synergies identified

Response 2: Thank you very much for your good comment. By comparing the trade-offs and synergies among ES obtained in this study with the existing research results, it is proved that it is reasonable, and relevant literature has been added in 4.2.

Point 3: Some comments on the temporal variations 2000-2019 within ES bundles.

Response 3: Thank you very much for your good comment. We re-analyzed Section 3.3, adding a comparison between 2000 and 2019. ES bundle 1 accounting for 69.91% of the total area in 2000 and 69.33% in 2019. Except for the decrease in grass supply from 2000 to 2019, all other ES have improved in bundle 1, namely food supply, soil conservation, carbon sequestration, SO2 absorption, NOx absorption, humidification, sand fixation, flood regulation, water conservation, outdoor recreation had a synergistic relationship, while they had a trade-off relationship with grass supply. ES bundle 2 accounting for 13.55% and 15.45% of total area in 2000 and 2019, respectively, with an increasing trend, mainly converted from bundle 5. Grass supply, NOx absorption, and sand fixation increased in 2019 compared to 2000, while other ES decreased, indicating that grass supply, NOx absorption, and sand fixation were synergies and trade-offs with food supply, soil conservation, carbon sequestration, SO2 absorption, humidification, flood regulation, water conservation, and outdoor recreation. ES bundle 3 accounting for 4.60% and 5.52% of total area, respectively. Water conservation contributed the most in 2000, while grass supply contributed the most in 2019. Only carbon sequestration increased from 2000 to 2019, while other ES decreased, that is, each ES in bundle 3 decreased with the increase of carbon sequestration, indicating that they have a trade-off relationship with carbon sequestration. ES bundle 4 accounting for 7.60% and 7.89% of total area in 2000 and 2019, respectively, compared with 2000, grass supply, soil conservation, humidification, sand fixa-tion, flood regulation and outdoor recreation have increased in 2019, while others have decreased, namely grass supply, soil conservation, humidification, sand fixation, flood regulation and outdoor recreation were synergistic, while food supply, carbon seques-tration, SO2 absorption, NOx absorption, water conservation were trade-offs. ES bundle 5 accounting for 4.33% of the total area in 2000 and 1.80% in 2019. Food supply, carbon sequestration, SO2 absorption, NOx absorption, humidification, sand fixation, and water conservation all improved in 2019 compared to 2000, while others have decreased, namely food supply, carbon sequestration, SO2 absorption, NOx absorption, humidification, sand fixation and water conservation are synergies, while they were trade-offs with grass supply, soil conservation, flood regulation and outdoor recreation.

Point 4: Furthermore, some very small language problems that can be solved by a thorough reading.

Response 4: Thank you very much for your good comment. We have read through the manuscript and corrected language problems.

Thank you very much for your comments and suggestions again.

Sincerely,

Shuang Gan

Reviewer 3 Report

The focus on tradeoffs and synergy between specific ES is certainly welcome, but the paper would become easier to follow if a land cover typology were used as a step between 'drivers' and the 'ES bundles', The specific definition of 'bundles' should be more clearly described from the introduction onwards.

The title could be shorter, e.g. something like "Ecosystem service bundle synergy and tradeoffs in Shenyang, China" 

Line 18 A few more words on the type of place Shenyang is can help the reader...

Line 33: Please start by defining ES ('benefits people derive from functioning (agro)ecosystems') -- which means that you can aggregate over people ('who benefits?') and types of benefits. Does the 'bundle' concept clarify who benefits how?

Line 37: How can identification of ES bundles change tradeoffs? It can make the tradeoffs more explicit...

Line 50: in a cascade model (structure ==> function ==> ES ==> benefits ==> people) one might expect 'ES bundles' to associate first with functin and then with structure...

Line 55: again -- do you approach this from the people/benefits side or from the structure/function angle?

Line 67 Please try to make a bridge between the generic interest in ES bundles that the first paragraphs describe, and the specific geographic context for the current paper? Why here? What can the reader expect? How relevant might this study be for other places?

Line 68-75 Instead of describing what the research team did, can you please formulate the questions you tried to address? What you did comes in section 2.

Line 91 Maybe mention fodder or animal feed instead of grass? 

Line 92 the 'sand fixation' needs a bit more explanation; is it really 'and' or control of wind erosion? dust bowls?

Line 92 What is the logic in the ordering of these 'services'? Maybe group the water-related services etc.

Line 92 "The specific calculation methods and 97
data sources are shown in Table 1." -- please do... a lot more detail is needed (maybe in an appendix or as supplementary material)

Table 1 As NPP is the basis for both 'food' and 'grass' -- how do you differentiate between the two?

Figure 1: the legends are too small to be readable; You have a lot of white space that could be used to convey information...

Line 164 The information in Table 1 on the operational definition of these ES is not enough to understand these results.

Figure 2: figures should be understandable through their caption -- please explain the many acronyms used

Figure 3 as for Fig. 2: caption incomplete

Line 197 Now the interpretation of 'ES bundles' appears to change -- as they primarily are pixels with a similar relative contribution to the various ES; they are not described on the basis of ecological structure and function that generate coherent ES categories, but spatial domains of similarity...

Line 220-234 This description would be easier to follow if the reader understands what land cover types are involved...

Author Response

Thank you for your comments concerning our manuscript. Those comments are all valuable and very helpful for revising and improving our paper, as well as the important guiding significance to our researches. We have studied comments carefully and have made correction which we hope meet with approval. The main corrections in the paper and the responds to the reviewer’s comments are as flowing:

Point 1: The focus on tradeoffs and synergy between specific ES is certainly welcome, but the paper would become easier to follow if a land cover typology were used as a step between 'drivers' and the 'ES bundles', The specific definition of 'bundles' should be more clearly described from the introduction onwards.

Response 1: Thank you very much for your good comment.

In this study, we used land cover typology as a part of the socio-ecological factors in the section of identification of ES bundles drivers to analyze the main driving factors that affect the formation of ES bundles.

The introduction part does have a lot of deficiencies. We have made a comprehensive revision to the Introduction, and added specific explanation of ES bundle. ES bundles are groups of ES that recur in space or time and have similar trade-offs and synergies. It is made up of multiple ES with similar supply levels, trade-offs and synergies. It is produced as a result of the combined effects of natural and socialeconomic factors, and it can characterize the characteristics of a human-land coupling complex system to some extent. Different types of ES bundles distinguish the human-land coupling complex system, allow the ecological environment problems that exist under different bundle types to be targeted identified, and the internal ES composition, trade-offs, and synergies to be grasped. Propose effective policy measures to avoid or reduce trade-offs based on the trade-offs and synergies, which will help im-prove the multiple supply of ES and maximize the social well-being of ES.

Point 2: The title could be shorter, e.g. something like "Ecosystem service bundle synergy and tradeoffs in Shenyang, China"

Response 2: Thank you very much for your good comment. We are very sorry, after discussion with the team, we decided not to revise the title.

Point 3: Line 18 A few more words on the type of place Shenyang is can help the reader...

Response 3: Thank you very much for your good comment. We have added a description of Shenyang in the Abstract and Introduction. Shenyang is a significant economic development and grain production area in Northeast China, with a diverse range of ES types. Since 2000, with the implementation of regional ecological restoration projects and the acceleration of urbanization, the disturbance of human activities in this area has continued to increase, and it is urgent to find management solutions that balance economic development and ecological protection.

Point 4: Line 33: Please start by defining ES ('benefits people derive from functioning (agro)ecosystems') -- which means that you can aggregate over people ('who benefits?') and types of benefits. Does the 'bundle' concept clarify who benefits how?

Response 4: Thank you very much for your good comment. The introduction part does have a lot of deficiencies. We have added the concept of ecosystem services and the specific explanation of ES bundle.

Ecosystem services (ES) are the benefits that humans derive from ecosystems, which include the material resources and services that ecosystems provide to human society for human survival and are closely related to human well-being. The study of ES provides a comprehensive perspective on ecosystem utilization and management, and it is an important tool for quantifying human-nature relationships. According to the Millennium Ecosystem Assessment, ES can be classified as provisioning services, regulating services, supporting services, and cultural services. A wide range of ES have complex trade-offs and synergies. For example, the improvement of some provi-sioning services is frequently accompanied by the reduction of regulating and cultural services. The current ecological and environmental problems are primarily mani-fested in the simplification of regional ES, which leads to the continuous deterioration of various ES capabilities and the trade-off relationship among various ES, and has a sig-nificant impact on human health and happiness. Effective management of multiple ecosystems, as well as incorporating them into landscape management decisions and policy making, necessitates a thorough understanding of the interrelationships among various ES.

ES bundles are groups of ES that recur in space or time and have similar trade-offs and synergies. It is made up of multiple ES with similar supply levels, trade-offs and synergies. It is produced as a result of the combined effects of natural and so-cial-economic factors, and it can characterize the characteristics of a human-land cou-pling complex system to some extent. Different types of ES bundles distinguish the human-land coupling complex system, allow the ecological environment problems that exist under different bundle types to be targeted identified, and the internal ES com-position, trade-offs, and synergies to be grasped. Propose effective policy measures to avoid or reduce trade-offs based on the trade-offs and synergies, which will help im-prove the multiple supply of ES and maximize the social well-being of ES.

Point 5: Line 37: How can identification of ES bundles change tradeoffs? It can make the tradeoffs more explicit...

Response 5: Thank you very much for your good comment. We have added an explanation of this issue in the Introduction. ES bundles are made up of multiple ES with similar supply levels, trade-offs and synergies. It is produced as a result of the combined effects of natural and social-economic factors, and it can characterize the characteristics of a human-land coupling complex system to some extent. Different types of ES bundles distinguish the human-land coupling complex system, allow the ecological environment problems that exist under different bundle types to be targeted identified, and the internal ES composition, trade-offs, and synergies to be grasped. Based on the trade-offs and synergies propose effective policy measures to avoid or reduce trade-offs.

Point 6: Line 50: in a cascade model (structure ==> function ==> ES ==> benefits ==> people) one might expect 'ES bundles' to associate first with functin and then with structure...

Response 6: Thank you very much for your good comment. The introduction part does have a lot of deficiencies. We have made a comprehensive revision to the Introduction. We mainly use the analysis of ES structure as a research method to achieve the purpose that people can sustainably obtain more benefits from the ecosystem.

Point 7: Line 55: again -- do you approach this from the people/benefits side or from the structure/function angle?

Response 7: Thank you very much for your good comment. The introduction part does have a lot of deficiencies. We have made a comprehensive revision to the Introduction. We mainly use the analysis of ES structure as a research method to achieve the purpose that people can sustainably obtain more benefits from the ecosystem.

Point 8: Line 67 Please try to make a bridge between the generic interest in ES bundles that the first paragraphs describe, and the specific geographic context for the current paper? Why here? What can the reader expect? How relevant might this study be for other places?

Response 8: Thank you very much for your good comment. We have completely revised the introduction based on your question.

Point 9: Line 68-75 Instead of describing what the research team did, can you please formulate the questions you tried to address? What you did comes in section 2.

Response 9: Thank you very much for your good comment. We have completely revised the introduction based on your question. This study conducts a dynamic analysis of the interrelationships and driving factors of various ES from the perspective of ES bundles, which is useful in providing a scientific reference for the formulation of relevant ecological protection policies.

Point 10: Line 91 Maybe mention fodder or animal feed instead of grass?

Response 10: Thank you very much for your good comment. Considering the computability and visualization of ES, we mainly consider the feed supply of cattle and sheep, and the feed of cattle and sheep is mainly grass, so we take grass supply as an indicator of provisioning service.

Point 11: Line 92 the 'sand fixation' needs a bit more explanation; is it really 'and' or control of wind erosion? dust bowls?

Response 11: Thank you very much for your good comment. Sand fixation mainly calculates the reduction of local wind erosion, and does not calculate the reduction of sand and dust outside the study area. We have added detailed calculation methods in the supplementary material.

Point 12: Line 92 What is the logic in the ordering of these 'services'? Maybe group the water-related services etc.

Response 12: Thank you very much for your good comment. We have added specific reasons for choosing 11 ES indicators in Material and methods. The selection was made in conjunction with existing literature after considering the types of ecosystems in Shenyang and their ecological, economic, and social functions, as well as the availability of data. Shenyang's land use type is primarily farmland, accounting for roughly 70% of total area. As a result, food supply is a significant ES in Shenyang. Shenyang also has forest land distribution in various regions, including deciduous broad-leaved forest, evergreen coniferous forests, deciduous coniferous forests, mixed coniferous and broad-leaved forests, deciduous broad-leaved shrub forests, arbor gardens, shrub gardens, arbor green spaces, etc., the vegetation types are relatively rich, the functions of SO2 absorption, NOx absorption, carbon sequestration, soil conservation, humidification are relatively prominent. Grassland is distributed in some areas, which can provide fodder for livestock and have the function of grass supply. Shenyang is in China's northeastern region, close to the source of sandstorms, and is part of the Three-North Shelter Forest Project. The sand fixation of forest is very important. The rivers are vertical and horizontal, with the Liaohe, Hunhe, Raoyanghe, and many other rivers, which have the functions of water conservation and flood regulation. The distribution of forests and rivers provides places for outdoor recreation. Finally, we chose 11 ES indicators to study ES bundles.

Point 13: Line 92 "The specific calculation methods and 97 data sources are shown in Table 1." -- please do... a lot more detail is needed (maybe in an appendix or as supplementary material)

Response 13: Thank you very much for your good comment. We have added detailed calculation methods and data sources in the supplementary material.

Point 14: Table 1 As NPP is the basis for both 'food' and 'grass' -- how do you differentiate between the two?

Response 14: Thank you very much for your good comment. We have added detailed calculation methods in the supplementary material. Food supply is calculated based on NPP and the actual grain output in Shenyang. Grass supply is calculated based on NPP, belowground to aboveground biomass ratio, standard hay conversion coefficient and grazing utilization.

Point 15: Figure 1: the legends are too small to be readable; You have a lot of white space that could be used to convey information...

Response 15: Thank you very much for your good comment. We have modified the legend.

Point 16: Line 164 The information in Table 1 on the operational definition of these ES is not enough to understand these results.

Response 16: Thank you very much for your good comment. We have added detailed calculation methods in the supplementary material. And we have added the unit of each indicator in Table 1.

Point 17: Figure 2: figures should be understandable through their caption -- please explain the many acronyms used

Response 17: Thank you very much for your good comment. We have added the full name of each acronyms to the caption of each figure.

Point 18: Figure 3 as for Fig. 2: caption incomplete

Response 18: Thank you very much for your good comment. We have added the full name of each acronyms to the caption of each figure.

Point 19: Line 197 Now the interpretation of 'ES bundles' appears to change -- as they primarily are pixels with a similar relative contribution to the various ES; they are not described on the basis of ecological structure and function that generate coherent ES categories, but spatial domains of similarity...

Response 19: Thank you very much for your good comment. We have added a specific explanation of ES bundles in the Introduction. ES in the same ES bundle have the same supply level and the same trade-offs and synergistic relationship. The ecological structure is mainly represented by how much each ES contributes and the trade-offs and synergies among ES. Figure 5 is normalized and drawn for each bundle before the modification. This result can only be compared to the ES within each bundle, and cannot be compared between bundles and between different years. Now, all ES in 2000 and 2019 are normalized together, and the rose diagram are drawn to make each bundle and each year comparable. We re-analyzed the results and discussions of ES bundle, and increased the  analyzation of internal trade-offs and synergies of each bundle.

Point 20: Line 220-234 This description would be easier to follow if the reader understands what land cover types are involved...

Response 20: Thank you very much for your good comment. We have added the analysis of land use types in the ES bundle discussion section and added the spatial distribution map of land use types in Shenyang in 2000 and 2019 in the supplementary material.

Thank you very much for your comments and suggestions again.

Sincerely,

Shuang Gan

Round 2

Reviewer 1 Report

Thank you very much for the revised manuscprit and for taking my comments thouroughly into account.

I have no further comments.

Reviewer 3 Report

Thanks for responding to reviewer suggestions